# Evaluation of the Bourban Trunk Muscle Strength Test Based on Electromyographic Parameters

**DOI:** 10.3390/jfmk4020035

**Published:** 2019-06-17

**Authors:** Stephan Becker, Joshua Berger, Marco Backfisch, Oliver Ludwig, Michael Fröhlich

**Affiliations:** Department of Sport Science, Technische Universität Kaiserslautern, 67663 Kaiserslautern, Germany

**Keywords:** electromyography, trunk muscle strength, assessment, fatigue, mean frequency, median frequency

## Abstract

(1) Background: The importance of a strong and stable trunk musculature is well known, but there is a lack of reliable, valid and objective test batteries with the necessary test economy, practicability and cost-benefit ratio. The aim of the present study was to evaluate the Bourban test for the validity of its exercise selection representing the ventral, right/left lateral and dorsal muscle chain. (2) Methods: 33 male sports students (24.1 ± 2.4 years, 181.6 ± 5.5 cm, 80.8 ± 7.3 kg) participated in the study. Median Frequency (MDF) and Mean Frequency (MNF) were calculated from the electromyographic signals and used to check whether fatigue of the suggested target musculature actually occurs during the different exercises and thus the exercise is representative for this part of the trunk. (3) Results: In all exercises significant fatigue was measured for MDF and MNF in the muscles working as agonists. (4) Conclusion: It can be stated that the Bourban trunk muscle strength test is a valid and economic test instrument for the evaluation of trunk strength (endurance). Compared to technically supported measuring systems, the Bourban test seems to be a flexible and cost-effective alternative for the broad mass.

## 1. Introduction

The trunk can be regarded as the anatomical basis of movements of the distal segments. Almost all large muscles, which are mainly responsible for basic movements, have their attachments here [1]. It allows the transfer of torques and angular momentum between lower and upper extremities with minimal loss of energy [2,3]. The following muscles can be listed as the main representatives of the trunk musculature: External oblique, internal oblique, transversus abdominis, rectus abdominis, transversospinalis, quadratus lumborum, erector spinae, latissimus dorsi, gluteus maximus and medius [2,4].

The importance of a well developed and stable trunk musculature is well known for everyday life [5], sports [1,6] and affects all age groups [7,8]. In everyday life, trunk stability supports general functionality [1], well-being [6] and is decisively involved in activities of daily life such as sitting, standing, walking or maintaining an upright posture [5]. Futhermore great importance is attributed to the prevention or treatment of back pain. Improved motor control and coordination contribute to an increased spinal stability [9]. From the point of view of sports science, there are implications ranging from injury prevention [10], general sports-specific fitness [11] to athletic performance improvement [12].

After trunk strengthening has become a sort of fitness trend and has influenced many sports. The question of a test instrument for the operationalisation of trunk strength arises at the latest when proof of effectiveness wants to be provided. Technological progress opens up many possibilities, ranging from mobile force sensors and devices to isokinetic force-measuring chairs. Unfortunately, these are associated with high acquisition costs and high technical proficiency requirements [6]. Up to now there is a lack of reliable, valid and objective test batteries with the necessary test economy, practicability and cost-benefit ratio, especially for screening procedures in everyday life, recreational sports, leisure sports, but also for professional sports [1,13].

The Bourban test is described as an economic alternative to other assessments measuring trunk strength endurance [14]. It can be applied in a pre-post-comparison and offers, even if so far as to be still limited for sport-specific normative values comparisons for the categorisation of trunk strength endurance [14,15]. The reliability of the test has already been proven [16].

The aim of the present study was to evaluate the Bourban test for the validity of its exercise selection representing the ventral muscle chain, right lateral muscle chain, left lateral muscle chain and dorsal muscle chain. According to our hypothesis, there should have been a significant decrease in median frequency (MDF) and mean frequency (MNF) from pre-test to post-test for the agonistic working muscles in all exercises.

## 2. Materials and Methods

### 2.1. Participants

A total of 33 male sports students (age: 24.1 ± 2.4 years, height: 181.6 ± 5.5 cm, weight: 80.8 ± 7.3 kg) participated in the study. Inclusion criteria were: male participants, between 20 and 30 years of age, previous experiences in strength training and with the basic forms of the exercises. Exclusion criteria were: acute injury, acute illness, acute muscle soreness or fatigue due to other sporting activities. All participants were informed in writing and orally about the study design, execution and potential risks and provided their written consent. Participation was voluntary and did not involve any financial remuneration. The study was based and carried out in accordance with the current guidelines of the Declaration of Helsinki [17] and approved by the responsible ethics commission of the Technische Universität Kaiserlautern.

### 2.2. Bourban Test

The Bourban test consists of four exercises, one for each part of the trunk: Ventral chain, right lateral chain, left lateral chain and dorsal chain (Figure 1, Figure 2, Figure 3 and Figure 4). All exercises were performed according to the authors’ guidelines [15,16] and to the point of subjective complete exhaustion or the exercises were stopped by the investigator if the range of motion or movement speed no longer met the guidelines. The movement speed was controlled by means of a metronome (60 bpm) and the range of motion was controlled by spanned ropes (Sport-Thieme^®^, Grasleben, Germany) [14]. If one of the specifications was deviated from, a warning was issued. The test was aborted at the third warning.

#### 2.2.1. Ventral Muscle Chain

The subjects took a front plank position with head, back, hips and legs forming a straight line (Figure 1). During the test, the subjects had to lift the straightened legs alternately to the level of the gluteus maximus. One leg had to be lifted at each beat of the metronome. The lower back, at the height of the sacroiliac joint, had to remain always in contact with the rope.

#### 2.2.2. Lateral Muscle Chain

The subjects took a side plank position with one hand on the hip (Figure 2 and Figure 3). Head, shoulders, back, and hips had to be positioned in one line. A pelvis drop and lift was performed in time. At each beat the subject had to be in the reversal point (top or bottom).

#### 2.2.3. Dorsal Muscle Chain

The subjects performed a hyper extension in a range of 0–30° according to the beat (Figure 4). The spinae iliacae anterior was about 4 cm above the edge and the arms were crossed over the breast. Head, back and hips had to form a straight line. At each beat, the trunk had to be in the reversal point (top or bottom).

Further details on the test setup, execution and instructions can be found in the existing literature [5,15,16,18].

### 2.3. Myoelectrical Activity

The electromyographic parameters MDF and MNF were used to check whether fatigue of the suggested target musculature actually occurred during the different exercises and thus the respective exercise could be representative for this part of the trunk. MDF and MNF are frequency parameters and can be used to determine local fatigue. With increasing fatigue the frequencies of the measured motor units will decrease and with MDF and MNF the magnitude of this shift can be shown [19].

The myoelectrical activity (Table 1, Figure 5 and Figure 6) during the four exercises (Ex.1–Ex.4) was determined by telemetric surface electromyography by Noraxon Desktop DTS (Noraxon, Scottdale, PA, USA; sampling frequency: 1500 Hz; filter: lowpass 500 Hz) according to the SENIAM standard [20]. Ag/AgCl adhesive electrodes (Ambu Blue Sensor P, 34 mm diameter, Ambub A/S, Ballerup, Denmark) were used to record the signals. The MDF and MNF were calculated at two points in time (pre-test, post-test) using the Fast-Fourier-Transformation in MR3 Software (MR3 Version 12.56, Noraxon, Scottdale, PA, USA). For the pre- and post-test, a standardized time window of 4 s was analyzed shortly after the start (pre-test) and shortly before termination (post-test) of the respective exercise and muscle.

### 2.4. Rate of Perceived Exertion (RPE) Scale

Before the start of the experiment, the participants were introduced to the use and interpretation of the Borg scale [21]. Immediately after the completion of each exercise, the rate of perceived exertion (RPE) was tested using the Borg scale (6 to 20 scale).

### 2.5. Statistics

The following results were stated as mean values ± standard deviation and 95% confidence intervals. To analyse the pre-post-effects the t-test for dependent samples was applied. The normal distribution was verified by means of the Shapiro–Wilk test. The significance level was modified by Bonferroni to: Ex.1: *p* < 0.016, Ex.2: *p* < 0.05, Ex.3: *p* < 0.05, Ex.4: *p* < 0.025. Effect sizes (Cohen’s d) were also calculated and values of 0.20, 0.50, and 0.80 were considered small, medium and large, respectively. The statistical evaluation was executed using IBM SPSS (SPSS Version 25.0 for Macintosh; Chicago, IL, USA).

## 3. Results

All respective muscles for Ex.1–Ex.4 showed a significant decrease in mean frequency (MNF) and median frequency (MDF). In Ex.1, representing the ventral muscle chain OEA_r, OEA_l and RAB MNF decreased on average by −12.1% and MDF −8.0%. The lateral muscle chain was tested in Ex.2 and Ex.3. Throughout Ex. 2 OEA_r decreased by −11.8% for MNF and −7.2% for MDF. A decrease of −12.3% for MNF and −5.9% for MDF was measured for OEA_l in Ex. 3. During Ex.4, representative for the dorsal muscle chain ESL and EST MNF decreased on average by −21.9% and MDF by −14.2%.

The following table shows the results of the pre-post-comparison for MNF and MDF in Hz (Table 2, Figure 7) as well as the descriptive statistics on duration and RPE scale for Exercise 1 (Ex.1)–Exercise 4 (Ex.4) (Table 3).

## 4. Discussion

The aim of the study was to investigate, if each of the four exercises in the Bourban trunk muscle strength test really represented the strength endurance of the ventral, lateral and dorsal muscle chain. In the case of a valid exercise selection, an electromyographically measurable decrease of the MNF and MDF of the agonistic muscles should occur in all exercises.

Tschopp et al. [18] already checked the validity of their test instrument with a questionnaire survey. The correctness of the operationalization of trunk strength with this test instrument was confirmed by 70% of the athletes (*n* = 57) and 100% of the coaches/sports instructors (*n* = 10). This subjective assessment can be extended and confirmed with our values of the RPE scale, which achieved an average of 16.6 to 17.0 for all exercises (Table 3). According to Borg’s RPE Scale, this corresponds to a maximal effort activity with very hard loads [21].

The statistical analysis of MNF and MDF of the respective muscles throughout the four exercises, showed significant fatigue values over time. Therefore the EMG frequency power spectrum shifted to lower frequencies during all exercises caused by local fatigue [19]. MNF and MDF for all muscles representing the ventral (OEA, RAB), lateral (OEA) and dorsal (ESL, EST) chain showed significantly lower frequencies in the post-test (Table 2). Tong et al. [22] evaluated the sport-specific endurance plank test on 34 participants with surface electromyography, supporting our results of Ex.1 challenging the target muscles of the ventral muscle chain RAB and OEA. Willardson et al. [23] investigated the myoelectrical activity of RAB, OEA, ESL and the lower abdominal stabilizers during the side plank (Ex.2 and Ex.3). According to their findings the agonist for the side plank was OEA and synergistically supported by RAB. Their results corroborated the Bourban test and confirmed that Ex.2 and Ex.3 must have led to fatigue in the lateral muscle chain. Similar to Ex.4, Yoshitake et al. [24] investigated the fatigue of ESL during isometric hyper extension over a period of 60 s. The EMG power spectrum, comparable to our results, shifted to lower frequency bands. In our study the average load duration was 86.6 s (Table 3). Looking at the available literature the usual object of investigation is the determination of muscular activity or the comparison of muscular activity during various exercises. Knowing that a certain muscle belongs to the respective agonists of the corresponding muscle chain and therefore experiences the greatest stress, an occuring fatigue seems trivial. In the current literature, there is a lack of studies which prove an objective fatigue with the help of electromyography for the exercises involving front plank, side plank and hyper extension. Nevertheless, according to the previous investigations for the Bourban test on the subjective [19] as well as the objective data available here, a valid measuring instrument can be assumed.

Looking at the exercise selection of the Bourban test more closely and comparing it with previously known tests, one can see that variants and variations have been used by other authors [13,25,26,27]. They all claim to be intended for the evaluation of trunk strength and differ in terms of content, structure and also the proportion of static and dynamic exercise elements [13,25,27,28]. The Bourban test can be understood as an advanced test of the already existing, simple dynamic tests, in which the measuring accuracy is increased with the help cost-effective and flexible variations (devices, test description, instructions, abort criteria) [15,16]. In a reliability test, 30 male athletes from 16 sports were tested at two test times within 48 h. The random error to be tested for this was 12–15% for the ventral (*r* = 0.87), lateral (*r* = 0.81) and dorsal chain (*r* = 0.80) [16]. An exact test description, instructions and standardization devices were used to ensure independence from the investigator [29]. A future study should evaluate the inter-tester reliability [16].

Apparative measurement methods usually provide reliable data, although they are usually too expensive and time-consuming for the broad mass [16]. Another weak point of apparatus-based measurement technology is that isometric or isokinetic measurement methods are often not representative of everyday motor skills or sports-specific performance. This is probably one of the reasons why some studies have not been able to sufficiently reveal evidence of efficacy and correlations between increased trunk strength and athletic performance [3]. Although the Bourban test already contains dynamic elements, which is altogether more intensive and more representative for sports practice [30], it is still more the static muscle endurance that is being tested [12]. Thus, athletes and coaches must always bear in mind that the primary goal is the determination of the basic strength or the effective control of a training intervention in the pre-post-comparison, but not necessarily a direct transfer to performance parameters in certain sports setting [28,31].

The comparative values available so far can be traced back to two surveys [14,15] and relate predominantly to male athletes (male: *n* = 598; female: *n* = 184). Future studies should therefore expand the inclusion criteria in order to collect comparative values for all age groups, sports groups and especially for women. According to the authors’ previous understanding [16], the primary objective is to categorize whether an athlete meets the minimum strength requirement or not, since a further increase is not necessarily associated with a proportional linear increase in athletic performance or preventive benefit [14]. Later Büsch et al. [15] started categorizing the performance into five stages. To answer the question on how to handle the data and categorize the perfomance, whether a division was sufficient or insufficient is adequate, or an extension according to the principle of Büsch et al. [15] is better, is to be discussed elsewhere. First of all, there is a need for an elementary increase in the number of comparative values.

Fatigue of the target musculature is guaranteed with the elements of the Bourban test, but it should be taken into account that other anatomical structures can have a limiting effect. Bourban et al. [14] were able to state that in the specification of the location of the main load the ventral chain in Ex.1 was 65.8%, the lateral chain in Ex.2 and Ex.3 was 84.0% and the dorsal chain in Ex.4 was 98.7% of the subjects. When testing the ventral chain, the lower back was mentioned as a possible limiting factor. The authors see here a radiating symptomatology by the overload of M. ilipsoas [14], because the activity of ESL or EST is clearly lower than that of EOA and RAB [32]. In our sample it could also be observed that M. quadriceps femoris could be perceived as a limiting factor. We explained the connection by the synergistically involved M. rectus femoris, since he is supporting the abdominal muscles with a slight hip flexion and working against a pelvic drop following gravity [33]. For this reason, attention should be paid to a recovered condition for the test, without any influencing strain having occurred in the training units of the previous days, also for possiblly supporting muscle groups. In addition, our observations coincided with those of Bourban et al. (2001) that the shoulder could be seen as a limiting factor for testing the lateral chain and the hamstrings for the dorsal chain [14].

As potential limitations of this study it can be noted that in our investigation only large, superficially lying and auxiliary muscles for the ventral, lateral and dorsal chain were recorded. An important contribution to trunk stability is made not only by the superficial muscles, but also by smaller and deeper lying muscles (e.g., M. iliopsoas, M. quadratus lumborum, M. obliquus internus abdominis, M. transversus abdominis, M. semispinalis) [34,35]. However, these can only be insufficiently or not at all recorded by means of surface electromyography. Nonetheless, due to their at least synergistic activation throughout these exercises, it can be assumed that adequate fatigue would also be measured for these muscles. Furthermore no female participants were included, which limits generalizability.

## 5. Conclusions

Based on our results, it can be stated that the Bourban trunk muscle strength test is a valid and economical test instrument for the evaluation of trunk strength endurance. In the exercises for the ventral, lateral and dorsal muscle chain, significant fatigue occured in the muscles working as agonists. The aim of the Bourban test was not to compete with technically supported measuring systems, but to represent a flexible and cost-effective alternative for the broad mass. It allows the determination and, if necessary, categorization of trunk strength, but does not replace tests for sport-specific performance parameters. Future studies should increase the comparative data for the subgroups age, gender, sport and untrained persons.

## Figures and Tables

**Figure 1 jfmk-04-00035-f001:**
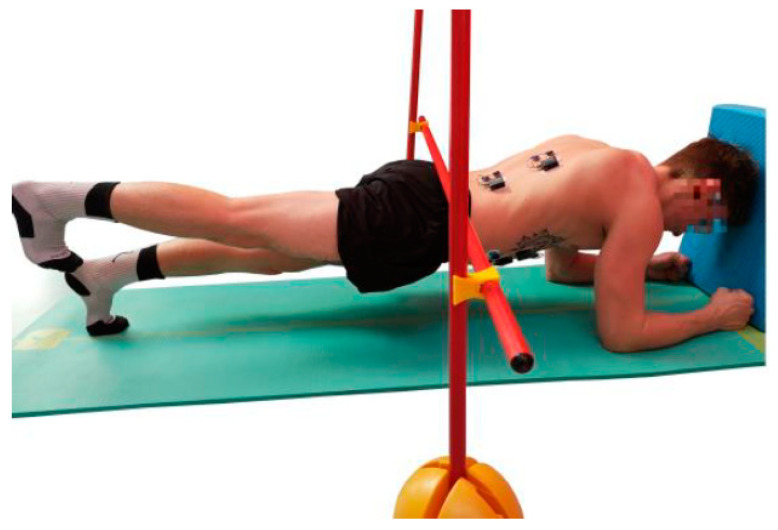
Exercise for the ventral muscle chain (plank with alternating leg lifts).

**Figure 2 jfmk-04-00035-f002:**
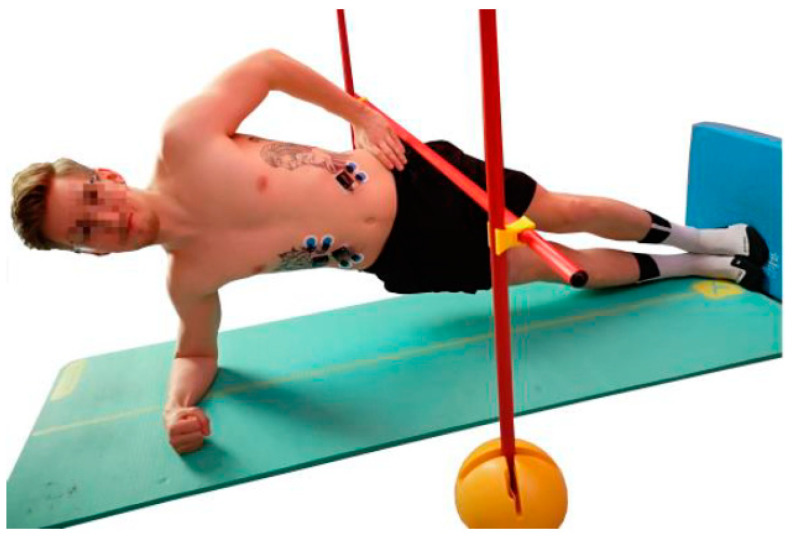
Exercise for the right lateral muscle chain (plank with pelvis drop and lift).

**Figure 3 jfmk-04-00035-f003:**
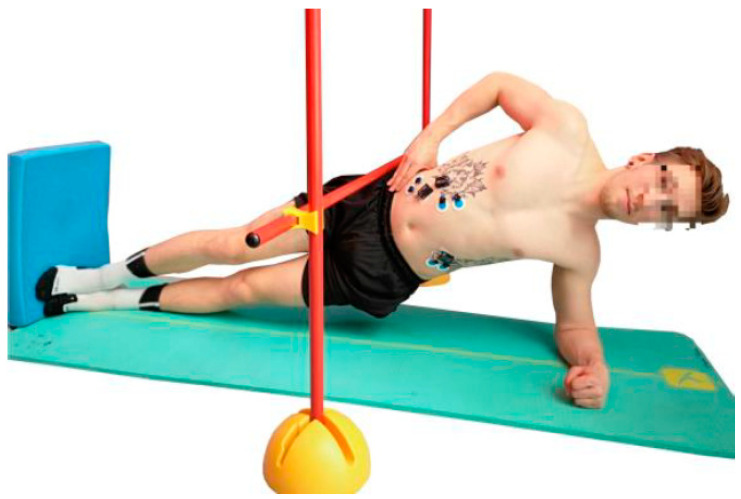
Exercise for the left lateral muscle chain (plank with pelvis drop and lift).

**Figure 4 jfmk-04-00035-f004:**
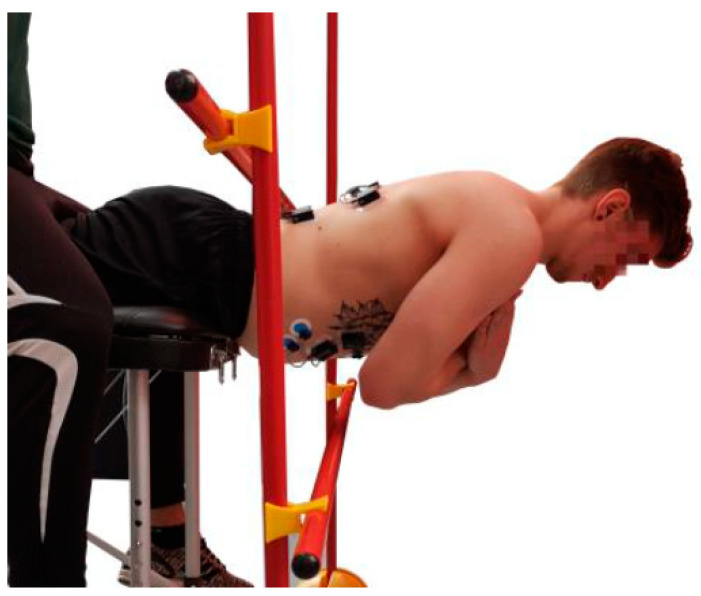
Exercise for the dorsal muscle chain (dynamic hyper extension).

**Figure 5 jfmk-04-00035-f005:**
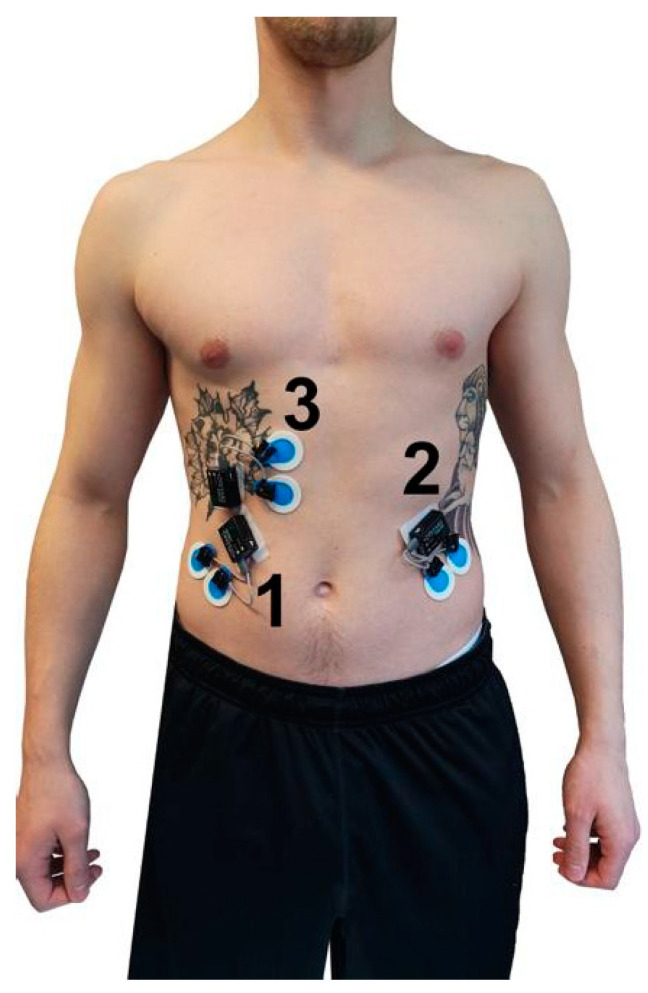
Position of the surface EMG electrodes for OEA_r (1), OEA_l (2) and RAB (3).

**Figure 6 jfmk-04-00035-f006:**
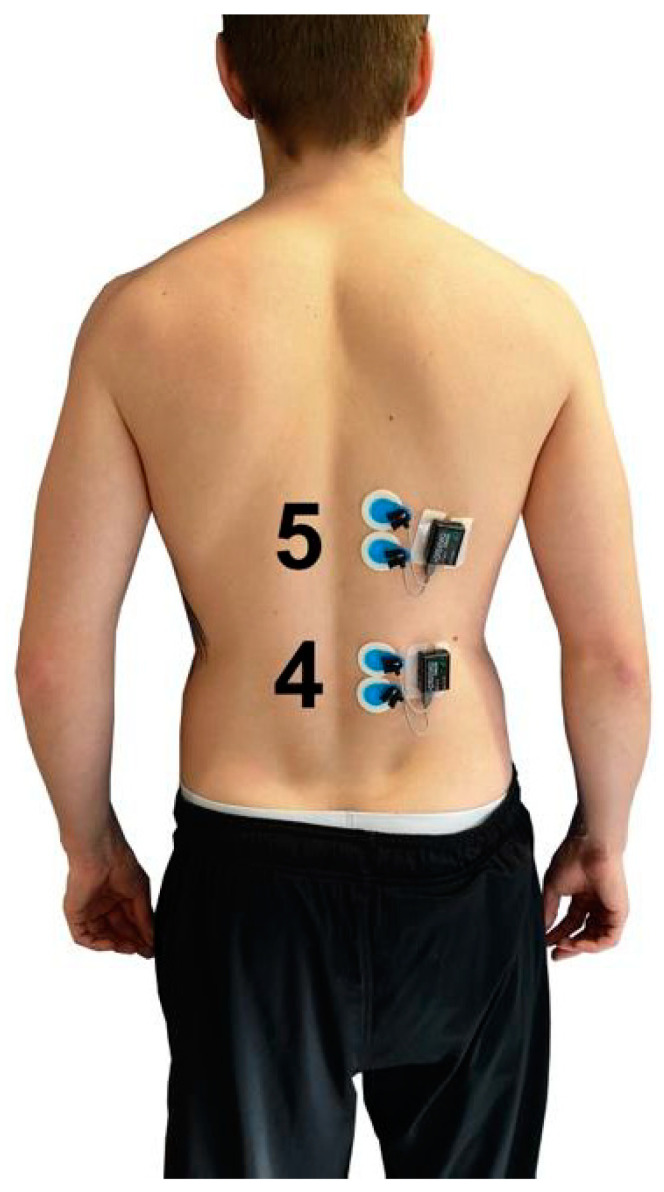
Position of the surface EMG electrodes for ESL (4) and EST (5).

**Figure 7 jfmk-04-00035-f007:**
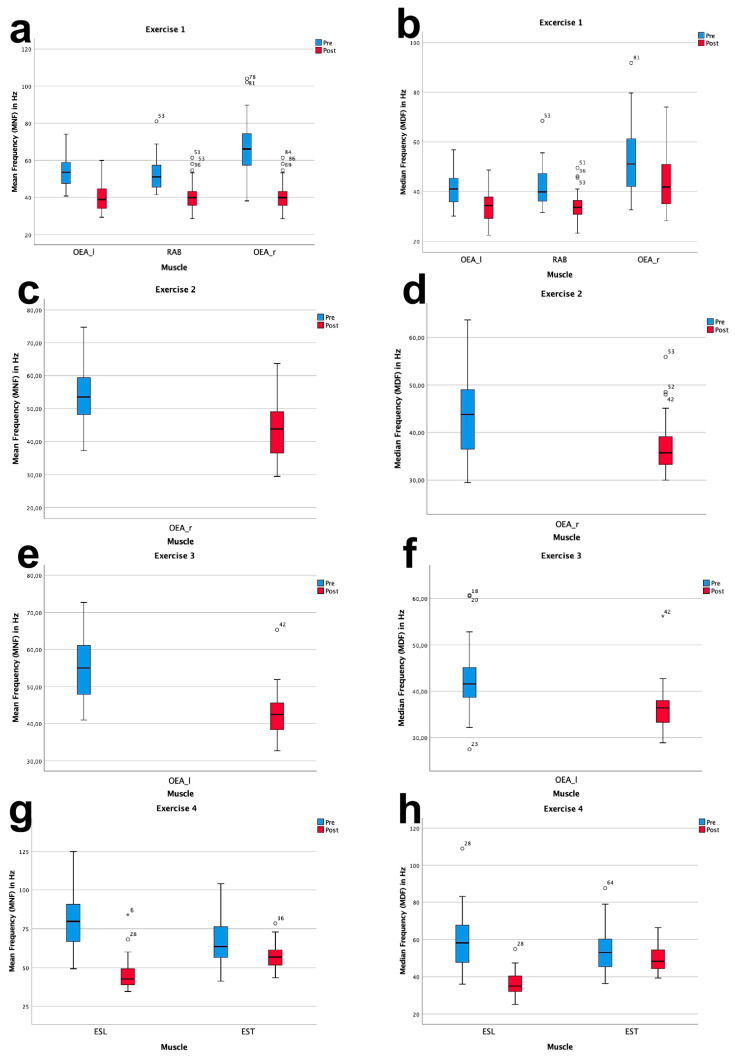
Boxplots showing the results for MNF (left column) and MDF (right column) during Ex.1 (**a**,**b**), Ex.2 (**c**,**d**), Ex.3 (**e**,**f**) and Ex.4 (**g**,**h**).

**Table 1 jfmk-04-00035-t001:** Ex.1–Ex.4 and the respective derived muscles.

Exercise	Muscle
Ex.1: ventral muscle chain	M. obliquus externus abdominis right-side (OEA_r) M. obliquus externs abdominis left-side (OEA_l) M. rectus abdominis (RAB)
Ex.2: lateral muscle chain (right-side)	M. obliquus externus abdominis right-side (OEA_r)
Ex.3: lateral muscle chain (left-side)	M. obliquus externus abdominis left-side (OEA_l)
Ex.4: dorsal muscle chain	M. erector spinae lumbalis (ESL) M. erector spinae thoracalis (EST)

**Table 2 jfmk-04-00035-t002:** Mean values of mean frequency (MNF in Hz) and median frequency (MDF in Hz) between pre- and post-test for exercises 1–4 (*N* = 33).

Ex.	Parameter	Pre ± SD	CI	*N*	Post ± SD	CI	T	df	*p*	*d*
1	OEA_r MNF	52.6 ± 8.8	49.4–55.7	33	40.7 ± 7.7	38.0–43.4	9.583	32	0.000	1.7
	OEA_r MDF	41.7 ± 8.0	38.8–44.5	33	34.2 ± 5.9	32.2–36.3	6.705	32	0.000	1.2
	OEA_l MNF	53.9 ± 8.1	51.0–56.8	33	40.5 ± 7.7	37.5–43.4	10.027	32	0.000	1.7
	OEA_l MDF	41.7 ± 6.7	39.3–44.0	33	33.9 ± 6.4	31.6–36.1	6.756	32	0.000	1.2
	RAB MNF	66.3 ± 15.6	60.8–71.8	33	55.3 ± 14.1	50.3–60.3	4.278	32	0.000	0.7
	RAB MDF	53.4 ± 18.8	48.6–58.3	33	44.6 ± 11.4	40.6–48.7	3.726	32	0.001	0.6
2	OEA_r MNF	54.7 ± 10.0	51.1–58.3	33	42.9 ± 7.6	40.3–45.6	7.614	32	0.000	1.3
	OEA_r MDF	44.3 ± 9.1	41.0–47.5	33	37.0 ± 6.0	34.9–39.1	4.518	32	0.000	0.8
3	OEA_l MNF	54.8 ± 8.7	51.7–57.9	33	42.5 ± 6.6	40.2–44.9	8.514	32	0.000	1.5
	OEA_l MDF	42.2 ± 7.5	39.6–44.9	33	36.3 ± 5.1	34.5–38.1	4.571	32	0.000	0.8
4	ESL MNF	80.8 ± 19.08	74.0–87.6	33	45.4 ± 10.5	41.6–49.1	10.866	32	0.000	1.9
	ESL MDF	59.3 ± 10.5	53.6–65.1	33	36.2 ± 6.3	34.0–38.4	8.784	32	0.000	1.5
	EST MNF	65.6 ± 14.0	60.6–70.6	33	57.1 ± 8.2	54.3–60.0	4.561	32	0.000	0.8
	EST MDF	55.0 ± 12.6	50.5–59.4	33	49.8 ± 7.3	47.2–52.4	3.012	32	0.005	0.5

Ex. = Exercise number (according to Table 1), SD = standard deviation, CI = 95% confidence interval, *N* = number of subjects, T = t-value, df = degrees of freedom, *p* = p-value, *d* = Cohen’s d effect size, OEA_r = M. obliquus externus abdominis (right-side), OEA_l = M. obliquus externus abdominis (left-side), RAB = M. rectus abdominis, ESL = M. erector spinae lumbalis, EST = M. erector spinae thoracalis.

**Table 3 jfmk-04-00035-t003:** Duration and RPE Scale of each exercise in average ± standard deviation.

Exercise	Duration in s	RPE Scale
1	151.2 ± 49.2	16.9 ± 1.7
2	62.0 ± 18.0	16.6 ± 2.3
3	64.9 ± 20.2	16.8 ± 1.9
4	86.6 ± 20.07	17.0 ± 1.8

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
