# Peer review of "Evaluation of the Bourban Trunk Muscle Strength Test Based on Electromyographic Parameters"

_jfmk, 2019, doi:10.3390/jfmk4020035_

Round 1

Reviewer 1 Report

General Comments

General Weaknesses 

- Authors must improve the myoelectrical activity procedures. They wrote that “the median and mean frecuency were calculated”. This explanation results confusing (they must explain how they are able to determine if tireness is expressed with this EMG analysis -or if tireness is not expressed). In addition, they must explain how these EMG parameters were calculated (lines 83-84, page 2 of 10).  

- Results section must be improved. I recommend to consult the journal´s guidelines. This section cannot be replaced for several tables (lines 132-146, pages 4 and 5 of 10).

- Discussion must be elaborated again. Authors must focus this section to disccuss their study´s results. They must compare their results with those of similar studies (which must be updated). In addition, they must try to elucite their study´s results underlying mechanisms. Finally, they must include the study´s limitations at the end of the section (the method-critical consideration presented could be used as a study´s limitation). I recommend to consult the journal´s guidelines to elaborate this section (pages 5 and 6 of 10).

General Strengths

- Statiscal analysis is correct and pertinent (lines 122-129, page 4 of 10).

Major Comments:

Title

Strengths

- Title is correct and pertinent (page 1 of 10).

Abstract

Weaknesses 

- Abstract must be improved following the present´s report recommendations (lines 18-30, page 1 of 10). 

Keywords

Weaknessess

- Authors must decrease keywords (I recommend to write 3-5 of them) (lines 31-33, page 1 of 10).

Introduction

Weaknesses 

- Introduction is correct and interesting (it is supported on pertinent, enough and updated quotations). Nevertheless, Authors must locate this part of the study´s protocol: “The electromyographic parameters Median Frequency (MDF) and Mean Frequency (MNF) [21] were used to check whether fatigue of the suggested target musculature actually occurs during the different exercises and thus the respective exercise is representative for this part of the trunk” (lines 64-67, page 2 of 10) on Material and Methods. In addition, Authors must present an overall study´s hypothesis, which must be located at the end of Introduction. 

Methods

Weaknesses 

- I recommend to use this word: ¨Participants” instead of this one: “ Subjects” (line 69, page 2 of 10).

- Why do Authors do not present the study´s inclusion criteria (lines 71-72, page 2 of 10). 

- I recommend to use these concepts: “myoelectrical activity” and “myoelectrical activity” instead of theses ones: “Electromiography” and “neuromuscular activity” (lines 78 and 79, page 2 of 10) (…myoelectrical activity was recorded…)

- Authors must improve the myoelectrical activity procedures. They wrote that “the median and mean frecuency were calculated”. This explanation results confusing (they must explain how they are able to determine if tireness is expressed with this EMG analysis -or if tireness is not expressed). In addition, they must explain how these EMG parameters were calculated (lines 83-84, page 2 of 10).  

- I recommend to use this order to present Methods: 2.2. Bourban test, 2.3. Myoelectrical activity and 2.4. RPE. It set the Bourban test as main focus of the section and the rest assessments as complentary ones (pages 2-3 of 10). 

Strengths

Statistical Analysis

- Statiscal analysis is correct and pertinent (lines 122-129, page 4 of 10).

Results

Weaknesses 

- Results section must be improved. I recommend to consult the journal´s guidelines. This section cannot be replaced for several tables (lines 132-146, pages 4 and 5 of 10).

Discussion

Weaknesses 

Discussion must be elaborated again. Authors must focus this section to disccuss their study´s results. They must compare their results with those of similar studies (which must be updated). In addition, they must try to elucite their study´s results underlying mechanisms. Finally, they must include the study´s limitations at the end of the section (the method-critical consideration presented could be used as a study´s limitation). I recommend to consult the journal´s guidelines to elaborate this section (pages 5 and 6 of 10). 

Conclusions

Weaknesses

- Despite that Conclusions are interesting, Authors must improve this section. They must delete quotations from this section (I recommend to consult the journal´s guideines) (pages 6 and 7 of 10). 

References

- This section must be checked it in details. It could contain format mistakes. In addition, number of references must be decrease (30-35 would be a pertinent number) (pages 9 and 10 of 10).    

Tables and Figures

Strengths

- Tables and Figures are correct and pertinent. They content a lot of interesting and useful data. However, they should be presented at the end of the manuscript (Auhtor should indicate places where they want tables are located with a sentence like this: ***Insert Table x or Figure x over here***) (pages 2, 3 4 and 5 of 10).

Author Response

Reviewer Nr. 1

Dear Reviewer,

thank you very much for the great effort and time you took. Your feedback and recommendations were completely justified. The current changes have already significantly improved the quality of our paper. If certain changes should not be satisfactory, then please give us a sign.

General Weaknesses

- Authors must improve the myoelectrical activity procedures. They wrote that “the median and mean frecuency were calculated”. This explanation results confusing (they must explain how they are able to determine if tireness is expressed with this EMG analysis -or if tireness is not expressed). In addition, they must explain how these EMG parameters were calculated (lines 83-84, page 2 of 10). 

·                     Good point. We added the missing informations. The Parameters were calculated with the MR3 Software which offers a fatigue report. This fatigue report is working with the Fast-Fourier-Transformation (page 3, lines 120-133)

- Results section must be improved. I recommend to consult the journal´s guidelines. This section cannot be replaced for several tables (lines 132-146, pages 4 and 5 of 10).

·                     We improved the results and added a further figure (page 3, line 152 – 251).

- Discussion must be elaborated again. Authors must focus this section to disccuss their study´s results. They must compare their results with those of similar studies (which must be updated). In addition, they must try to elucite their study´s results underlying mechanisms. Finally, they must include the study´s limitations at the end of the section (the method-critical consideration presented could be used as a study´s limitation). I recommend to consult the journal´s guidelines to elaborate this section (pages 5 and 6 of 10).

·                     Thank you for pointing out this missing part (page 4, lines 272-293).

General Strengths

- Statiscal analysis is correct and pertinent (lines 122-129, page 4 of 10).

Major Comments:

Title

Strengths

- Title is correct and pertinent (page 1 of 10).

Abstract

Weaknesses

- Abstract must be improved following the present´s report recommendations (lines 18-30, page 1 of 10).

·                     We hopefully implemented all your recommendations to a satisfactory degree.

Keywords

Weaknessess

- Authors must decrease keywords (I recommend to write 3-5 of them) (lines 31-33, page 1 of 10).

·                     Now 5: Trunk stability has been deleted as an possible key word. (3-10 key words are allowed)

Introduction

Weaknesses

- Introduction is correct and interesting (it is supported on pertinent, enough and updated quotations). Nevertheless, Authors must locate this part of the study´s protocol: “The electromyographic parameters Median Frequency (MDF) and Mean Frequency (MNF) [21] were used to check whether fatigue of the suggested target musculature actually occurs during the different exercises and thus the respective exercise is representative for this part of the trunk” (lines 64-67, page 2 of 10) on Material and Methods. In addition, Authors must present an overall study´s hypothesis, which must be located at the end of Introduction.

·                     The sentence has been moved to materials and methods.

·                     Our hypothesis is now clearly formulated at the end of the introduction

Methods

Weaknesses

- I recommend to use this word: ¨Participants” instead of this one: “ Subjects” (line 69, page 2 of 10).

·                     See page 2, line 70

- Why do Authors do not present the study´s inclusion criteria (lines 71-72, page 2 of 10).

·                     Inclusion criteria have been added (page 2, lines 72-73)

- I recommend to use these concepts: “myoelectrical activity” and “myoelectrical activity” instead of theses ones: “Electromiography” and “neuromuscular activity” (lines 78 and 79, page 2 of 10) (…myoelectrical activity was recorded…)

·                     See: page 2, line 126; page 4, line 279

- Authors must improve the myoelectrical activity procedures. They wrote that “the median and mean frecuency were calculated”. This explanation results confusing (they must explain how they are able to determine if tireness is expressed with this EMG analysis -or if tireness is not expressed). In addition, they must explain how these EMG parameters were calculated (lines 83-84, page 2 of 10). 

·                     Good point. We added the missing informations. The Parameters were calculated with the MR3 Software which offers a fatigue report. This fatigue report is working with the Fast-Fourier-Transformation (page 3, lines 120-133)

- I recommend to use this order to present Methods: 2.2. Bourban test, 2.3. Myoelectrical activity and 2.4. RPE. It set the Bourban test as main focus of the section and the rest assessments as complentary ones (pages 2-3 of 10).

·                     Thank you for this recommendation. We have changed the order of the chapters (pages 2-3).

Statistical Analysis

- Statiscal analysis is correct and pertinent (lines 122-129, page 4 of 10).

Results

Weaknesses

- Results section must be improved. I recommend to consult the journal´s guidelines. This section cannot be replaced for several tables (lines 132-146, pages 4 and 5 of 10).

·                     We consulted the journal’s guidelines again and improved the results and added a further figure (page 3, line 152 – 251).

Discussion

Weaknesses

Discussion must be elaborated again. Authors must focus this section to disccuss their study´s results. They must compare their results with those of similar studies (which must be updated). In addition, they must try to elucite their study´s results underlying mechanisms. Finally, they must include the study´s limitations at the end of the section (the method-critical consideration presented could be used as a study´s limitation). I recommend to consult the journal´s guidelines to elaborate this section (pages 5 and 6 of 10).

·                     Thank you for pointing out this missing part (page 4, lines 272-293).

·                     Study’s limitations are now clearly visible at the end of the chapter (page 5, lines 375-383)

Conclusions

Weaknesses

- Despite that Conclusions are interesting, Authors must improve this section. They must delete quotations from this section (I recommend to consult the journal´s guideines) (pages 6 and 7 of 10).

·                     We deleted the quotations (page 6, lines 393-400)

References

- This section must be checked it in details. It could contain format mistakes. In addition, number of references must be decrease (30-35 would be a pertinent number) (pages 9 and 10 of 10).   

·                     This sections has been checked again.

·                     Number of references has been reduced to 35.

Tables and Figures

Strengths

- Tables and Figures are correct and pertinent. They content a lot of interesting and useful data. However, they should be presented at the end of the manuscript (Auhtor should indicate places where they want tables are located with a sentence like this: ***Insert Table x or Figure x over here***) (pages 2, 3 4 and 5 of 10).

·                     The changed the manuscript following your recommendations.

Reviewer 2 Report

The purpose of the present study was to evaluate the validity of the Bourban test for assessing the ventral, dorsal, and lateral trunk muscles. Thirty-three active males participated in the study and performed four trunk exercises for the ventral, dorsal, and right/left lateral musculature while EMG was recorded from the relevant agonist muscles in each posture. The myoelectric manifestations of muscle fatigue were quantified using the MDF and MNF, which were the main dependent variables.  It was hypothesized that the agonist muscle in each posture would fatigue and therefore MDF and MNF, which are closely related to motor unit conduction velocity, would decrease during the fatiguing contractions. Subjects performed the exercises to a metronome to muscular failure. The main findings were that the MDF and MNF decreased in the agonist muscles for a given posture. Thus, it was concluded that the Bourban test is a valid test of trunk muscle endurance and that it has several practical advantages over other methods.

Overall, the manuscript is well-written and extends previous research by other research groups who developed the Bourban test, but apparently have not performed EMG measurements concurrent with the test. The study design is simple, appropriate,  and the data appear to have been collected carefully. In addition, the authors also did a good job of explaining the results of their study in light of previous findings and of identifying the limitations of the study. The topic addressed is not extremely novel as it is pretty unsurprising that when an agonist muscle is activated over a long period of time that it fatigues and MNF and MDF decrease, but nonetheless this hasn't been done in this particular test. The focus of the research seems to be appropriate for the Journal of Functional Morphology and Kinesiology and will be of interest to many readers of the journal. There appear to be no major flaws in the paper and I do not have concerns with the study design and interpretation of results. I only have a few minor comments, additions, and typographical errors that the authors should address before the manuscript can be considered ready for publication.

Minor Points:

1.  Line 114 – Shouldn't "Ventral Muscle Chain" read "Dorsal Muscle Chain"?

2.  Line 202 – Shouldn't "quadripecs" read "quadriceps"?

3. I think it would be ideal to have a Figure of the MNF and MDF results.

4. Line 37 - Shouldn't "attachement" read "attachments"?

5. Line 50 – I would change "lots of" to "many" or something like that.

6. Line 54 – "know how" is somewhat of a jargon term I would say "high technical proficiency requirments" or something similar.

7. Some readers may like to know the name or brand name of the devices used in Figures 3-6 that consist of 3 poles and anchors. Readers of this journal may be interested in performing these exercises themselves, with patients, and/or clients. Thus, they may be interested in knowing the name of this equipment so that they could put these exercises in practice and perform the correct range of motion, adjust heights for various people, etc.

Author Response

Reviewer Nr. 2

Dear Reviewer,

thank you very much for your recommendations. Your feedback was completely understandable and right. If certain changes should not be satisfactory, then please give us a sign.

The purpose of the present study was to evaluate the validity of the Bourban test for assessing the ventral, dorsal, and lateral trunk muscles. Thirty-three active males participated in the study and performed four trunk exercises for the ventral, dorsal, and right/left lateral musculature while EMG was recorded from the relevant agonist muscles in each posture. The myoelectric manifestations of muscle fatigue were quantified using the MDF and MNF, which were the main dependent variables.  It was hypothesized that the agonist muscle in each posture would fatigue and therefore MDF and MNF, which are closely related to motor unit conduction velocity, would decrease during the fatiguing contractions. Subjects performed the exercises to a metronome to muscular failure. The main findings were that the MDF and MNF decreased in the agonist muscles for a given posture. Thus, it was concluded that the Bourban test is a valid test of trunk muscle endurance and that it has several practical advantages over other methods.

Overall, the manuscript is well-written and extends previous research by other research groups who developed the Bourban test, but apparently have not performed EMG measurements concurrent with the test. The study design is simple, appropriate,  and the data appear to have been collected carefully. In addition, the authors also did a good job of explaining the results of their study in light of previous findings and of identifying the limitations of the study. The topic addressed is not extremely novel as it is pretty unsurprising that when an agonist muscle is activated over a long period of time that it fatigues and MNF and MDF decrease, but nonetheless this hasn't been done in this particular test. The focus of the research seems to be appropriate for the Journal of Functional Morphology and Kinesiology and will be of interest to many readers of the journal. There appear to be no major flaws in the paper and I do not have concerns with the study design and interpretation of results. I only have a few minor comments, additions, and typographical errors that the authors should address before the manuscript can be considered ready for publication.

Thank you!

Minor Points:

1.  Line 114 – Shouldn't "Ventral Muscle Chain" read "Dorsal Muscle Chain"?

·                     You are absolutley right. Thank you for the hint (line 112).

2.  Line 202 – Shouldn't "quadripecs" read "quadriceps"?

·                     Typing error has been corrected (line 367)

3. I think it would be ideal to have a Figure of the MNF and MDF results.

·                     We added the figure you asked for (Figure 7).

4. Line 37 - Shouldn't "attachement" read "attachments"?

·                      This error has been corrected (line 37)

5. Line 50 – I would change "lots of" to "many" or something like that.

·                     Good point. Thank you.

6. Line 54 – "know how" is somewhat of a jargon term I would say "high technical proficiency requirments" or something similar.

·                     We endorsed your proposal (line 56)

7. Some readers may like to know the name or brand name of the devices used in Figures 3-6 that consist of 3 poles and anchors. Readers of this journal may be interested in performing these exercises themselves, with patients, and/or clients. Thus, they may be interested in knowing the name of this equipment so that they could put these exercises in practice and perform the correct range of motion, adjust heights for various people, etc.

·                     I have added some informations on the manufacturer (line 86).

Reviewer 3 Report

Manuscript is well written. However, some minor flaws need to be addressed:

L58: "The Bourban test is described as an economic alternative." Alternative to what? Please explain.

L 87: MR3 is NOT from Calgary! But it is a Noraxon product!

Figures: Please anonymise the faces exposed.

Ethical approval obtained by "responsible ethics committee" Where obtained? Do you have an ethical approval number? If yes, you have to state this.

Do not use "Tab.", but "Table"

Table 2 is not clear: Abbreviations not explained (N, CI, T, df, d). Furthermore, it is not displaying "Statistical mean differences" but only means.

L 213 : use the correct nomina anatomica!

Author Response

Reviewer Nr. 3

Dear Reviewer,

thank you very much for your recommendations. Your feedback was completely understandable and right. If certain changes should not be satisfactory, then please give us a sign.

Manuscript is well written. However, some minor flaws need to be addressed:

L58: "The Bourban test is described as an economic alternative." Alternative to what? Please explain.

·                     The added some information to clearify (line 60).

L 87: MR3 is NOT from Calgary! But it is a Noraxon product!

·                     Good point! Sorry for that mistake! (line 131)

Figures: Please anonymise the faces exposed.

·                     Faces have been anonymised.

Ethical approval obtained by "responsible ethics committee" Where obtained? Do you have an ethical approval number? If yes, you have to state this.

·                     Further informations have been added. Since there is no approval number, a official paper has been forwarded to the editorial board.

Do not use "Tab.", but "Table"

·                     Ok, thank you!

Table 2 is not clear: Abbreviations not explained (N, CI, T, df, d). Furthermore, it is not displaying "Statistical mean differences" but only means.

·                     Abbreviations do now have an explanation.

·                     Statistical mean differences has been corrected.

L 213 : use the correct nomina anatomica!

·                     Absolutley right (line 378)

Round 2

Reviewer 1 Report

No comments